# The Effect of Healthy Lifestyle Changes on Work Ability and Mental Health Symptoms: A Randomized Controlled Trial

**DOI:** 10.3390/ijerph192013206

**Published:** 2022-10-13

**Authors:** Rahman Shiri, Ari Väänänen, Pauliina Mattila-Holappa, Krista Kauppi, Patrik Borg

**Affiliations:** 1Finnish Institute of Occupational Health, Työterveyslaitos, P.O. Box 40, FI-00032 Helsinki, Finland; 2Aisti Health Ltd., FI-00120 Helsinki, Finland

**Keywords:** anxiety, depression, diet, exercise, sleep, work ability

## Abstract

Objective: The effects of lifestyle interventions on the prevention of a decline in work ability and mental health are not well known. The aim of this randomized controlled trial was to examine the effects of healthy lifestyle changes on work ability, sleep, and mental health. Methods: Workers aged 18–65 years, who were free from cardiovascular diseases, diabetes, and malignant diseases, and did not use medication for obesity or lipids were included (N = 319). Based on their cholesterol balance, participants were classified into medium-risk and high-risk groups and were randomized into four arms: group lifestyle coaching (N = 107), individual lifestyle coaching (N = 53), the control group for group coaching (N = 106), and the control group for individual coaching (N = 53). The intervention groups received eight sessions of mostly remote coaching for 8 weeks about healthy diet, physical activity, other lifestyle habits, and sources/management of stress and sleep problems, and the control groups received no intervention. In individual coaching, the coach focused more on individual problem solving and the possibilities for motivation and change. The intention-to-treat principle was applied, and missing data on the outcomes were imputed using multiple imputation. Results: After the completion of the intervention, the risk of depressive symptoms was lower by 53% (95% CI 1–77%) in participants who received individual lifestyle coaching compared with the control group. The intervention had no beneficial effects on anxiety, work ability, sleep duration, or daily stress. In subgroup analyses, group lifestyle coaching had beneficial effects on depressive symptoms and work ability in participants with less tight schedules or less stretching work, whereas individual lifestyle coaching lowered the risk of depressive symptoms in those with fewer overlapping jobs, less tight schedules, or less stretching work. Conclusion: Short but intensive remote lifestyle coaching can reduce depressive symptoms and improve work ability, and time-related resources at work may improve mental health in the context of individual lifestyle intervention. However, further randomized controlled trials are needed to confirm the findings.

## 1. Introduction

The modification of individuals’ lifestyle behaviors is at the core of chronic disease prevention. Healthy eating and physical activity reduce excess body mass, systolic and diastolic blood pressure, low-density lipoprotein cholesterol, triglycerides, and blood glucose [1]. There is a general agreement that lifestyle interventions require an individualized approach. Typically, lifestyle coaching is based on behavioral change theories [2,3]. For the maintenance of lifestyle change, individuals need intrinsic motivation, self-regulation skills, resources, and habit-forming skills [2,3]. Lifestyle coaching aims to help individuals make a healthy lifestyle change within these domains and maintain the change.

The beneficial effects of lifestyle interventions on health-related behavior and health indicators have been shown. A lifestyle intervention consisting of an app-based diet, exercise self-logging, and personalized coaching by dieticians and exercise coordinators was effective in reducing body weight and body fat mass; however, the intervention did not reduce blood pressure [4]. An individual lifestyle coaching intervention, along with access to e-health support (Heathesteps app, a private social network and telephone coaching), increased step counts, decreased sitting time, and improved healthy eating, but did not change physical activity, weight, waist circumference, blood pressure, or health-related quality of life [5]. Lifestyle coaching consisting of a comprehensive nutrition and health education program for individuals diagnosed as prediabetic or at risk of diabetes reduced body mass index and waist circumference but did not change systolic or diastolic blood pressure, glucose, cholesterol, or triglycerides [6]. A systematic review [7] found that healthy lifestyles reduced sickness absence, and exercise improved work ability and mental well-being.

Although the primary impact of lifestyle coaching is often focused on physical health due to intervention-related improvements, it may also have beneficial impacts on work ability, psychological well-being, and other dimensions of health and disability. To date, little is known about the effects of lifestyle changes on the prevention of work disability and mental health problems. However, positive results have been shown in some subgroups of the population. Meta-analyses of randomized controlled trials have found that exercise [8] and dietary improvements [9] have beneficial effects on depression. Low levels of physical activity were also associated with anxiety [10]. A randomized controlled trial found that group strength exercises during working hours, plus group motivational coaching, improved work ability among female healthcare workers [11]. Another randomized controlled trial showed that among workers with chronic illnesses, coaching to manage challenges and strains caused by chronic disease improved work ability and exhaustion burnout [12]. Moreover, healthy dietary coaching protected against major depression in at-risk older adults [13].

The aim of the current randomized controlled trial was to examine the effects of lifestyle coaching on work ability, mental health, and sleep.

## 2. Methods

### 2.1. Participants

All employees from willing companies that were customers of Aava Medical occupational health care were invited to take part in the study. The company representative delivered information about the study to all the company’s employees. Inclusion criteria were (1) healthy workers aged 18–65 years; (2) able to speak Finnish or English; (3) male or female, with at least a third of each gender in the company; and (4) signed written informed consent. The exclusion criteria were: (1) history of a major cardiovascular event (myocardial infarction, coronary artery bypass grafting, percutaneous coronary intervention, stroke, or transient ischemic heart attack) during the preceding six months; (2) receiving treatment for type 1 or type 2 diabetes; (3) history of a malignant disease during the preceding five years; (4) using a lipid-lowering medication; (5) using medication for obesity; (6) using a cardiac pacemaker or history of atrial fibrillation; (7) pregnancy; and (8) having a plan to travel more than a day per week during the trial period. All participants fulfilling the eligibility criteria were invited to the screening phase. The trial was completed in June 2021.

### 2.2. Randomization

The eligible participants were randomized using a block randomization generated by a computer. Since the primary objective of this trial was to ascertain the effects of lifestyle coaching on blood metabolite levels, participants were classified into high-risk and medium-risk groups based on their cholesterol balance, a proxy for cardiometabolic risk, and the ratio of serum apolipoprotein B to apolipoprotein A1 concentrations were used to measure the proportion of serum atherogenic (mainly LDL) and anti-atherogenic (HDL) lipoprotein particles [14]. Participants within the highest 1/3 of ApoB/ApoA1 ratios were defined as the high-risk group and randomized into groups receiving personal lifestyle coaching or control group. Participants within the lower 2/3 of ApoB/ApoA1 ratios were defined as the medium-risk group and randomized into groups receiving group lifestyle coaching or control group. In this trial, neither allocation group was concealed nor were participants or caregivers blinded.

### 2.3. Intervention

Participants in both intervention groups received a structured lifestyle coaching program by educated and experienced coaches consisting of eight mostly remote (due to COVID-19) sessions within a period of 8 weeks (1 h every week). Participants in the control group had access to usual occupational health services and received no intervention. During individual coaching, the coach focused more on individual problem solving, feedback, and possibilities for motivation and change. During group coaching, 3 to 8 individuals took part in group coaching, and participants received peer support and peer-based problem solving, but there were fewer possibilities for individual focus. The coaching topics were similar in individual and group coaching and included four basic domains: (1) healthy nutrition; (2) physical activity; (3) other lifestyle changes; and (4) stress management and sleep. The aim was to increase intake of fruits, vegetables, and unsaturated fats; reduce alcohol consumption, salt, and sugar; increase leisure time physical activity to recommended levels of 150–300 min of moderate intensity activity or 75–100 min of vigorous intensity activity per week; and to identify sources of stress and sleep problems and manage them. The theoretical background of the intervention lay in the theory of behavioral change [2,3]. Group intervention also utilized the idea of peer support.

### 2.4. Outcomes

The Whooley questions were used to assess the risk of depression [15] and included two questions from the Primary Care Evaluation of Mental Disorders (PRIME-MD) Patient Health Questionnaire (PHQ) [16]. The questions were: (1) during the past month, have you often been bothered by feeling down, depressed, or hopeless?; and (2) during the past month, have you often been bothered by little interest or pleasure in doing things? [15]. Anxiety was assessed using the Generalized Anxiety Disorder 2-item (GAD-2) [17]. The two questions were: (1) in the past two weeks, I have been bothered by feeling nervous, anxious, or on edge; and (2) in the last two weeks, I have not been able to stop or control worrying. Additionally, a visual analog scale (0 to 100) was used for the responses. We used the average score of the two questions as a continuous variable. In addition, we also used it as a binary outcome and defined being at risk of anxiety as having a score above 80.

Work ability was assessed using the Work Ability Index [18]. The questionnaire consists of seven questions about current work ability and its relation to job demands, number of health conditions, work impairment due to health conditions, number of sickness absences, prognosis of work ability, and mental resources. The Work Ability Index scores range between 7 and 49. Sleep duration was assessed with a single question: “how many hours do you sleep on weekdays in general?”. Firstbeat Bodyguard 2 [19], a chest-strap wearable device, was used to assess daily stress and sleep duration. Firstbeat Bodyguard 2 measured heart rate and activity for three consecutive days. The participants also recorded their sleeping and waking times. The recorded beat-to-beat and motion data were transformed into health indices reflecting physical activity, stress, and recovery. Based on heart rate variability, when autonomic nervous system sympathetic activity dominates over parasympathetic activity, the device detects it as stress.

### 2.5. Baseline Characteristics

The questionnaire elicited information on age, gender, level of education, and current smoking status. Height and weight were measured. For job strain, information on overlapping jobs, tight schedules, and stretching workdays was gathered using a visual analog scale (0 to 100). The use of this data to measure job strain can be justified by a study population [20]. The questions were: (1) I have too many overlapping jobs at work; (2) my work includes tight schedules; and (3) my workdays often stretch.

### 2.6. Statistical Analysis

The baseline characteristics of the participants in intervention and control groups were compared using percentages (%) and mean ± SD. The continuous outcome variables were not normally distributed. To compare the groups with respect to the baseline characteristics, *χ*^2^ tests were used for categorical variables, and the Kruskal–Wallis test was used for continuous variables. A total of 115 participants had missing data for one of the outcomes of the study. We applied the intention-to-treat principle, conducted a complete case analysis, and imputed missing data on the outcomes of the study using multiple imputation. The imputation model included outcomes and baseline covariates, and missing data were imputed separately for each randomized group [21,22]. We created 50 multiple imputed datasets using chained equations (“mi impute chained” in Stata) and performed an intention-to-treat analysis. Number of events, percentages, risk ratios (RR), and mean differences across 50 datasets were combined to produce one set of statistics (“mi estimate” in Stata). We used a generalized linear model with a robust variance estimator, and for binary outcomes, we specified a binomial distribution and a log link function. Furthermore, we conducted subgroup analyses to determine whether the effects of lifestyle coaching differ between participants who had low or high overlapping jobs, tight schedules, or stretched workdays. We used median distribution to classify participants into low or high levels.

## 3. Results

A total of 713 participants were screened for eligibility (Figure 1). Of these, 319 participants were randomized into four arms: group lifestyle coaching (N = 107), individual lifestyle coaching (N = 53), the control group for group lifestyle coaching (N = 106), and the control group for individual lifestyle coaching (N = 53). Information on at least one of the outcomes was available for 97 in group lifestyle coaching, 49 in individual lifestyle coaching, 91 participants in the control group for group lifestyle coaching, and 48 in the control group for individual lifestyle coaching. A total of 115 participants had missing data on at least one of the secondary outcomes of the study. In total, 34 participants had missing data on depressive and anxiety symptoms, 41 participants had missing data on work ability, and 88 had missing data on sleep duration and daily stress at the follow-up. The distribution of age, gender, education, body mass index, work ability, depression, and anxiety at baseline did not differ between participants with missing data for one or more outcomes at follow-up and those without missing data.

Of the randomized participants, 259 were men, and 62 were women. The mean age was 47.6 ± 8.4 years at the time of trial completion. The intervention arms did not differ with regard to age, education, smoking, depression, anxiety, work ability, sleep duration, or daily stress at baseline (Table 1). However, there were a few more women in the control group of individual lifestyle coaching, and the body mass index was higher in the control group than the intervention group received individual lifestyle coaching.

After intervention, the body mass index significantly reduced in the participants who received individual lifestyle coaching (*p* = 0.028), and the association was significant (*p* = 0.025) only in participants who reported depressive symptoms at baseline. Exercise increased in those who received group lifestyle coaching (*p* = 0.056), and the association was found (*p* = 0.075) only in participants who did not report depressive symptoms at baseline. No changes were observed in smoking and alcohol consumption.

Compared with the control group, the risk of depressive symptoms was lower in participants who received individual lifestyle coaching by 54% (95% CI 3–78%, risk ratio 0.46, 95% CI 0.22–0.97) using complete case analysis and by 53% (95% CI 1–77%, risk ratio 0.47, 95% CI 0.23–0.99) using imputed data (Table 2). There were no statistically significant differences in anxiety (yes/no outcome, Table 2; continuous outcome, Table 3), work ability, sleep duration, or daily stress between the intervention and control groups (Table 3). Sleep duration, measured using Firstbeat Bodyguard 2, did not differ between the intervention arms either.

Table 4 and Table 5 show the results of imputed data for the effects of lifestyle coaching on depressive symptoms, anxiety, work ability, sleep duration, and daily stress among participants with low or high overlapping jobs, tight schedules, and stretched work. Group lifestyle coaching reduced the risk of depressive symptoms and improved work ability among participants with less tight schedules or fewer stretched workdays. Individual lifestyle coaching reduced the risk of depressive symptoms among participants with fewer overlapping jobs, less tight schedules, or less stretching work. However, individual lifestyle coaching increased the risk of anxiety among participants with fewer overlapping jobs or less stretching work.

## 4. Discussion

The current randomized controlled trial suggests that individual lifestyle interventions can protect against depressive symptoms but do not improve anxiety, work ability, or sleep. However, in individuals with better time-related resources at work, healthy lifestyle changes not only reduced the risk of depressive symptoms but also improved work ability.

In the current trial, lifestyle changes did not have beneficial effects on anxiety or sleep. However, earlier randomized controlled trials found that at least 150 min weekly of moderate or vigorous leisure-time physical activity could reduce depression, anxiety, and insomnia symptoms [23], and an energy-restricted low-fat diet could improve depressive and anxiety symptoms in overweight and obese individuals [24]. Earlier studies showed the beneficial effects of lifestyle interventions on anxiety and sleep in longer follow-up periods (6–12 months) [23,24]. In a randomized controlled trial [25], individuals who reduced their leisure-time physical activity or increased sedentary behavior for a week were at higher risk of depressive symptoms compared with individuals who maintained their normal physical activity for a week, but they were not at increased risk of anxiety. However, lifestyle interventions aimed at increasing physical activity and healthy eating reduced depressive symptoms among overweight men at both 3 and 9 months [26]. One of the reasons for the absence of effects on anxiety and sleep in the current trial might be due to the short follow-up period.

In the current trial, excess body mass was reduced at follow-up in participants who reported depressive symptoms at the trial entry, while exercise increased in those free from depressive symptoms. There is some evidence that depressive symptoms do not interfere with lifestyle interventions. In a cohort study, depressive symptoms at baseline did not influence lifestyle changes at follow-up in patients with coronary artery disease [27]. A lifestyle intervention that consisted of adopting a healthy diet, increasing physical activity, and stress management improved depressive symptoms and quality of life in individuals with major depression [28]. In individuals with bipolar disorder, lifestyle interventions improved depressive symptoms, weight, physical activity, and serum lipids [29]. However, individual counseling aimed at reducing weight and increasing physical activity in middle-aged overweight or obese individuals with impaired glucose tolerance was not effective in reducing depressive symptoms [30].

Considerably little is known about the impact of work characteristics on the successfulness of lifestyle interventions on work-related and mental health-related outcomes. The current randomized controlled trial indicated that the impact of the lifestyle interventions was mainly found in employees with fewer overlapping jobs, less tight schedules, or less stretching work. The findings suggest that employees need sufficient time and other resources to benefit from lifestyle interventions organized by employers. This observation in the work-related setting is novel and is in line with earlier observations on the role of personal resources as a factor for lifestyle changes [3]. Time constraints are considered barriers to maintaining a healthy lifestyle [31]. To reach optimal results, it is advisable to modify employees’ resources at work so that they are able to benefit from coaching and other methods used in lifestyle interventions.

The coaches of the current trial had experience in lifestyle counseling in occupational settings, including group and individual coaching. A pilot study was conducted to assess the coaching effectiveness, and participant feedback was gathered. The pilot study showed that the coaching worked and that the participants’ experience was good. The current randomized controlled trial had some limitations. First, there was no follow-up, and only the short-term effects of lifestyle coaching on mental health, sleep, and work ability were studied. Second, neither allocation group was concealed, nor were the participants or caregivers blinded. Trials with inadequate allocation concealment, particularly those with subjective outcomes, may overestimate an intervention’s effect [32]. However, in the current trial, there were no significant imbalances between the baseline characteristics and the intervention arms. Recruiting a few more women or overweight/obese participants into the control group of individual lifestyle coaching could be attributed to chance, as the trial recruited a small number of participants into individual lifestyle coaching and its control group. Third, multiple outcomes were studied, and multiple subgroup analyses were conducted. None of the observed associations were statistically significant when the *p* values were controlled for multiple testing. Fourth, most of the outcomes were subjectively assessed. Knowledge of interventions may have influenced the assessment outcomes. For instance, participants in the intervention groups may have underestimated their risk of depressive symptoms. Finally, in the current trial, the Whooley questions were used to assess depressive symptoms. The Whooley questions have higher sensitivity than the PHQ-2. The time frame is longer than the PHQ-2 (last month vs. last 2 weeks), and the questions are constructed in a yes/no format. However, its specificity is lower than the PHQ-2.

## 5. Conclusions

Although the current randomized controlled trial had some methodological shortcomings, it offers novel longitudinal evidence on the plausible beneficial effects of lifestyle coaching on some aspects of mental health (lower levels of depressive symptoms) and work ability. Furthermore, from the perspective of arranging this kind of intervention, it indicates an important direction by suggesting that sufficient time-related resources at work may result in intervention success. However, further randomized controlled trials with sufficient statistical power, objectively measured outcomes, and a longer follow-up period are needed.

## Figures and Tables

**Figure 1 ijerph-19-13206-f001:**
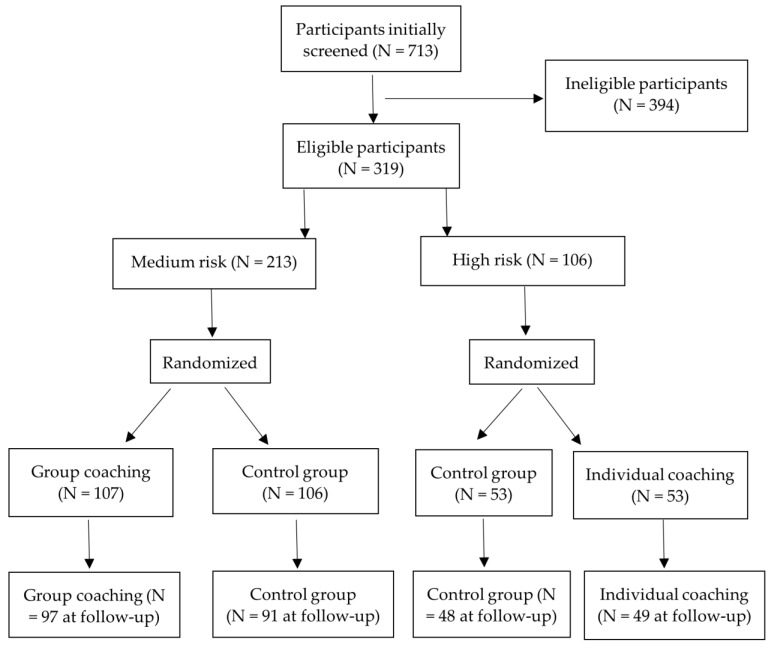
Flow chart of the study population.

**Table 1 ijerph-19-13206-t001:** The baseline characteristics of the intervention and control groups.

Characteristic	Control		Group Lifestyle Coaching		*p*	Control		Individual Lifestyle Coaching		*p*
	N	%	N	%		N	%	N	%	
Gender										
Men	79	74.5	86	80.4		43	81.1	49	92.4	
Women	27	25.5	21	19.6	0.30	10	18.9	4	7.6	0.085
Education										
No vocational school	3	2.8	2	1.9		0	0	3	5.7	
Vocational school	6	5.7	7	6.5		3	5.7	6	11.3	
College	42	39.6	48	44.9		24	45.3	24	45.3	
University	55	51.9	50	46.7	0.82	26	49.0	20	37.7	0.18
Smoking										
None	96	90.6	94	88.7		49	92.4	43	81.2	
Occasional	8	7.5	7	6.6		2	3.8	5	9.4	
Current	2	1.9	5	4.7	0.50	2	3.8	5	9.4	0.22
Depression	44/105	41.9	41/106	38.7	0.63	18/53	34.0	17/53	32.1	0.83
Anxiety	40/105	38.1	48/106	45.3	0.29	18/53	34.0	17/53	32.1	0.83
Company										
1	9	8.5	10	9.3		5	9.4	5	9.4	
2	87	82.1	86	80.4		42	79.3	42	79.3	
3	10	9.4	11	10.3	0.95	6	11.3	6	11.3	1.00
		**Mean (SD)**		**Mean (SD)**					**Mean (SD)**	
Age (years)	106	48.8 (8.2)	107	47.3 (8.3)	0.16	53	46.2 (7.9)	53	47.5 (9.3)	0.44
Body mass index	106	28.0 (4.3)	107	27.7 (5.5)	0.17	53	30.4 (5.6)	53	28.1 (3.8)	0.060
Work ability	105	41.5 (4.6)	105	40.8 (5.3)	0.51	53	40.7 (5.1)	52	41.5 (4.0)	0.42
Anxiety	105	69.5 (22.3)	106	71.2 (21.8)	0.63	53	71.9 (19.8)	53	71.1 (18.4)	0.52
Sleep duration (minutes)	82	469 (56)	83	468 (73)	0.61	38	448 (87)	39	464 (47)	0.58
Daily stress (minutes)	82	717 (211)	83	737 (184)	0.85	38	767 (168)	39	806 (156)	0.29

**Table 2 ijerph-19-13206-t002:** The effects of group and individual lifestyle coaching on depression and anxiety (binary outcomes) based on complete case analysis and multiple imputed data. RR, risk ratio; %, proportion of outcome.

Outcome	Complete Case Analysis						Imputed Data					
	N	Events	%	RR	95% CI	*p*	N	Events	%	RR	95% CI	*p*
Depression												
Control	91	36	39.6	1			106	42	39.6	1		
Group lifestyle coaching	97	30	30.9	0.78	0.53–1.16	0.219	107	32	29.9	0.78	0.52–1.15	0.205
Control	48	17	35.4	1			53	18	34.0	1		
Individual lifestyle coaching	49	8	16.3	0.46	0.22–0.97	0.041	53	9	17.0	0.47	0.23–0.99	0.049
Anxiety												
Control	91	39	42.9	1			106	45	42.5	1		
Group lifestyle coaching	97	39	40.2	0.94	0.67–1.32	0.713	107	42	39.3	0.94	0.67–1.32	0.726
Control	48	17	35.4				53	21	39.6	1		
Individual lifestyle coaching	49	24	49.0	1.38	0.86–2.23	0.185	53	26	49.1	1.36	0.85–2.18	0.205

**Table 3 ijerph-19-13206-t003:** The effects of group and individual lifestyle coaching on work ability, anxiety, sleep duration, and daily stress (continuous outcomes).

Outcome	Complete Case Analysis						Imputed Data					
	N	Mean	SD	Difference	95% CI	*p*	N	Mean	SD	Difference	95% CI	*p*
Work ability												
Control	87	41.6	5.5				106	41.6	5.6			
Group lifestyle coaching	97	42.4	4.7	0.83	−0.65, 2.31	0.27	107	42.2	4.8	0.57	−0.88, 2.02	0.44
Control	45	41.3	4.3				53	41.5	4.1			
Individual lifestyle coaching	49	42.2	5.1	0.91	−0.98, 2.80	0.34	53	42.3	5.3	0.62	−1.23, 2.48	0.51
Anxiety												
Control	91	73.1	21.7				106	73.2	21.0			
Group lifestyle coaching	97	71.6	22.2	−1.54	−7.80, 4.72	0.63	107	71.3	22.8	−1.73	−7.96, 4.49	0.58
Control	48	69.5	23.6				53	69.3	23.0			
Individual lifestyle coaching	49	76.4	20.0	6.92	−1.75, 15.58	0.11	53	75.7	21.0	6.97	−1.84, 15.78	0.12
Sleep duration												
Control	77	455	54				106	455	58			
Group lifestyle coaching	77	464	42	9.4	−5.7, 24.5	0.22	107	460	44	8.97	−6.77, 24.72	0.26
Control	38	453	53				53	457	54			
Individual lifestyle coaching	39	458	44	5.5	−16.0, 27.1	0.61	53	452	42	2.76	−20.85, 26.37	0.81
Daily stress (minutes)												
Control	77	760	218				106	734	221			
Group lifestyle coaching	77	780	154	20.3	−39.2, 79.8	0.50	107	792	155	15.5	−43.8, 74.8	0.60
Control	38	748	186				53	738	217			
Individual lifestyle coaching	39	755	185	6.9	−75.5, 89.3	0.86	53	757	203	−1.0	−94.3, 92.3	0.98

**Table 4 ijerph-19-13206-t004:** The results of imputed data on the effects of lifestyle coaching on depression and anxiety among participants with low or high overlapping jobs, tight schedules, and stretching work.

Type of Work	Depression			Anxiety		
	RR	95% CI	*p*	RR	95% CI	*p*
Overlapping job, low						
Control	1			1		
Group lifestyle coaching	0.74	0.45–1.20	0.222	1.03	0.55–1.95	0.920
Control	1			1		
Individual lifestyle coaching	0.26	0.08–0.85	0.026	2.11	1.06–4.20	0.034
Overlapping job, high						
Control	1			1		
Group lifestyle coaching	0.80	0.42–1.50	0.480	0.94	0.63–1.39	0.756
Control	1			1		
Individual lifestyle coaching	0.70	0.23–2.11	0.524	0.85	0.42–1.74	0.660
Tight schedules, low						
Control	1			1		
Group lifestyle coaching	0.56	0.31–1.01	0.056	1.04	0.63–1.74	0.874
Control	1			1		
Individual lifestyle coaching	0.35	0.11–1.12	0.076	1.49	0.72–3.07	0.282
Tight schedules, high						
Control	1			1		
Group lifestyle coaching	0.99	0.59–1.67	0.975	0.93	0.58–1.47	0.745
Control	1			1		
Individual lifestyle coaching	0.51	0.17–1.47	0.212	1.32	0.71–2.44	0.384
Stretch work, low						
Control	1			1		
Group lifestyle coaching	0.46	0.24–0.89	0.021	1.15	0.62–2.16	0.655
Control	1			1		
Individual lifestyle coaching	0.23	0.05–0.93	0.040	1.72	0.93–3.17	0.081
Stretch work, high						
Control	1					
Group lifestyle coaching	1.07	0.65–1.76	0.790	0.91	0.61–1.36	0.638
Control	1			1		
Individual lifestyle coaching	0.69	0.26–1.82	0.451	0.98	0.46–2.11	0.967

**Table 5 ijerph-19-13206-t005:** The results of imputed data on the effects of lifestyle coaching on work ability, anxiety, sleep duration, and daily stress among participants with low or high overlapping jobs, tight schedules, and stretching work.

Type of work	Work Ability			Anxiety			Sleep Duration			Daily Stress		
	Difference	95% CI	*p*	Difference	95% CI	*p*	Difference	95% CI	*p*	Difference	95% CI	*p*
Overlapping job, low												
Group lifestyle coaching	1.70	−0.49, 3.88	0.128	−2.25	−11.83, 7.32	0.644	8.39	−14.42, 31.21	0.471	−35.68	−118.52, 47.16	0.398
Individual lifestyle coaching	1.52	−0.87, 3.92	0.213	13.89	0.35, 27.43	0.044	1.32	−30.39, 33.03	0.935	−6.37	−131.65, 118.91	0.920
Overlapping job, high												
Group lifestyle coaching	−0.38	−2.26, 1.51	0.694	−0.97	−8.76, 6.83	0.808	7.81	−14.44, 30.05	0.491	64.01	−17.08, 145.11	0.122
Individual lifestyle coaching	−0.49	−3.42, 2.43	0.742	−0.59	−11.15, 9.97	0.913	4.62	−29.09, 38.33	0.788	5.81	−123.59, 135.21	0.930
Tight schedules, low												
Group lifestyle coaching	2.23	0.07–4.39	0.043	0.60	−8.21, 9.40	0.895	9.83	−12.35, 32.00	0.385	11.70	−73.99, 97.40	0.789
Individual lifestyle coaching	0.07	−2.75, 2.88	0.964	4.33	−10.16, 18.82	0.558	6.19	−28.40, 40.79	0.725	−0.39	−140.18, 139.40	0.996
Tight schedules, high												
Group lifestyle coaching	−0.93	−2.94, 1.07	0.360	−3.39	−12.30, 5.52	0.456	6.27	−16.12, 28.66	0.582	19.29	−61.65, 100.23	0.640
Individual coaching	0.97	−1.51, 3.45	0.443	8.95	−2.34, 20.24	0.120	0.49	−30.15, 31.14	0.975	−1.38	−122.69, 119.93	0.982
Stretch work, low												
Group lifestyle coaching	3.08	0.74, 5.42	0.010	1.12	−8.49, 10.74	0.819	9.39	−15.63, 34.40	0.462	−4.79	−99.57, 89.99	0.921
Individual coaching	1.28	−0.83, 3.40	0.235	14.76	4.22, 25.30	0.006	−5.52	−34.22, 23.18	0.706	−31.17	−149.75, 87.41	0.606
Stretch work, high												
Group lifestyle coaching	−1.13	−2.96, 0.71	0.229	−3.18	−11.32, 4.96	0.444	7.39	−12.17, 26.96	0.458	30.08	−45.66, 105.82	0.436
Individual lifestyle coaching	−0.25	−3.58, 3.08	0.883	−3.82	−18.30, 10.66	0.605	16.66	−21.41, 54.73	0.390	42.95	−94.89, 180.79	0.541

## Data Availability

The data presented in this study are available on request from the corresponding author.

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
