# Peer review of "The Effect of Healthy Lifestyle Changes on Work Ability and Mental Health Symptoms: A Randomized Controlled Trial"

_ijerph, 2022, doi:10.3390/ijerph192013206_

Round 1

Reviewer 1 Report

The current work "The Effect of Healthy Lifestyle Changes on Workability and Wellbeing: A Randomized Controlled Trial" by Shiri et al. is generally interesting and shows novelty. This reviewer has some comments:

Major Comments:

* The authors concentrate on cardiovascular risk factors in screening, group allocation and intervention, but on mental health, workability and sleep in outcomes. There needs to be a more elaborate background and discussion linking these domains for a reader to follow the rationale of the study.

* Again, there was an allocation to high risk and low risk groups in the methods and it appears they did not matter for the study’s analyses or results. Was this study part of a bigger project? Please elaborate further.

* Apparently the baseline characteristics were assessed using a self-made tool? Why was specifically information on “overlapping jobs”, “tight schedules” and “stretching workdays” evaluated for the job strain assessment?

Minor Comments:

* Abstract: “prevention of workability” (line 9)? -> “prevention of decrease in workability” maybe?

* Introduction: “Lifestyle coaching aims to help individuals with these domains to make a healthy lifestyle change and maintain the change.” (line 43) -> “Lifestyle coaching aims to help individuals to make a healthy lifestyle change within these domains and maintain the change.”?

* Methods: “a chest-strap wearable device, was used to assess daily stress and sleep duration” (line 141) Please further elaborate on how “stress” is measured here; HRV? Please elaborate on the values used and their interpretation in this specific context

*Statistical Analysis: “A total of 115 participants had missing data on one of the out-comes of the study. Thirty-four participants had missing data on depressive and anxiety symptoms, 41 participants had missing data on workability and 88 had missing data on sleep duration and daily stress at the follow-up.” These are results and should be moved to the Results section.

* Discussion: While the authors appear to provide an extensive literature research much of their findings are not well explained. E.g. Second paragraph of the discussion: the authors found no beneficial effects of their intervention on anxiety and sleep, but other authors did before. How do the authors explain the lack of effect? Specifically for sleep, which was targeted by their interventions.

*Table 2: Please include more information in the table description. The reader should be able to understand a table based on its description alone.

* Minor language issues throughout the manuscript

e.g. “A lifestyle coaching consisted of a comprehensive nutrition and health educational program […] reduced body mass index and waist circumference” (line 54 in Introduction) -> “consisting of” etc.

Author Response

The current work "The Effect of Healthy Lifestyle Changes on Workability and Wellbeing: A Randomized Controlled Trial" by Shiri et al. is generally interesting and shows novelty. This reviewer has some comments:

 Response: Thank you for your comments!

Major Comments:

* The authors concentrate on cardiovascular risk factors in screening, group allocation and intervention, but on mental health, workability and sleep in outcomes. There needs to be a more elaborate background and discussion linking these domains for a reader to follow the rationale of the study.

Response: The primary objective of this trial was to study the effect of lifestyle coaching on blood metabolite levels. Cholesterol metabolism, blood glucose, triglycerides, low grade inflammation, amino acid metabolism and purine metabolism were measured. The secondary objective was to study the effect of lifestyle coaching on workability, mental health symptoms and sleep. We have now explained this on page 3, 2nd paragraph.

* Again, there was an allocation to high risk and low risk groups in the methods and it appears they did not matter for the study’s analyses or results. Was this study part of a bigger project? Please elaborate further.

Response: A block randomization was used to randomize the participants into four arms to balance the intervention and control groups for the level of serum cholesterol. The results differed between medium- and high-risk groups. Thus, we did not combine medium- and high-risk groups into one group. This study was not part of a bigger project. Workability, mental health, and sleep were part of the trial’s secondary outcomes.

* Apparently the baseline characteristics were assessed using a self-made tool? Why was specifically information on “overlapping jobs”, “tight schedules” and “stretching workdays” evaluated for the job strain assessment?

Response: For the job strain, only information on overlapping jobs, tight schedules and stretching workdays was collected. The use of those items in job strain measure can be justified by a study population. Most of the participants worked in knowledge jobs. Overlapping jobs, tight schedules and stretching workdays are typical in these jobs. https://link.springer.com/article/10.1007/s41542-020-00058-1

Minor Comments:

* Abstract: “prevention of workability” (line 9)? -> “prevention of decrease in workability” maybe?

Response: Thank you for noticing this mistake. We have modified it as suggested.

* Introduction: “Lifestyle coaching aims to help individuals with these domains to make a healthy lifestyle change and maintain the change.” (line 43) -> “Lifestyle coaching aims to help individuals to make a healthy lifestyle change within these domains and maintain the change.”?

Response: We have modified the statement on page 2 as suggested.

* Methods: “a chest-strap wearable device, was used to assess daily stress and sleep duration” (line 141) Please further elaborate on how “stress” is measured here; HRV? Please elaborate on the values used and their interpretation in this specific context

Response: We have added the following explanation to the methods section on page 4, first paragraph.

“Based on heart rate variability, when autonomic nervous system sympathetic activity dominates over parasympathetic activity, the device detects it as stress.

*Statistical Analysis: “A total of 115 participants had missing data on one of the out-comes of the study. Thirty-four participants had missing data on depressive and anxiety symptoms, 41 participants had missing data on workability and 88 had missing data on sleep duration and daily stress at the follow-up.” These are results and should be moved to the Results section.

Response: We have moved those results to the results section on page 4 as suggested.

* Discussion: While the authors appear to provide an extensive literature research much of their findings are not well explained. E.g. Second paragraph of the discussion: the authors found no beneficial effects of their intervention on anxiety and sleep, but other authors did before. How do the authors explain the lack of effect? Specifically for sleep, which was targeted by their interventions.

Response:  We have added the following explanation to the second paragraph of the discussion.

“One of the reasons for the absence of effects on anxiety and sleep in the current trial might be due to short follow-up period.”

These results of earlier studies are already described in the second paragraph of the discussion as follows: “earlier studies showed beneficial effects of lifestyle interventions on anxiety and sleep in a longer follow-up period (6-12 months) (Brinkworth et al., 2009, Hartescu et al., 2015).” 

*Table 2: Please include more information in the table description. The reader should be able to understand a table based on its description alone.

Response: We have revised the title of Table 2 to explain better the results.

* Minor language issues throughout the manuscript
e.g. “A lifestyle coaching consisted of a comprehensive nutrition and health educational program […] reduced body mass index and waist circumference” (line 54 in Introduction) -> “consisting of” etc.

Response: We have performed language editing.

Reviewer 2 Report

The article is very interesting and presents well-planned research. I have 3 comments on how to present the results:

1. The authors mention that the studies were conducted during Covid-19. Could this circumstance and possible changes in the course of examinations of the individual health condition or health of family members modify the obtained results, especially those concerning the level of anxiety, depressive symptoms, and daily stress?

2. I would suggest a clearer comparison of the effects of individual and group lifestyle coaching as a guide when planning future research.

3. The authors point out that the lifestyle changes demonstrating the success of the intervention were mainly seen among workers with fewer overlapping tasks, less busy schedules, or less work-intensive work. Hence, it is correct to conclude that it is advisable to modify the resources of employees at work so that they can benefit from coaching and other methods used in lifestyle interventions. I would suggest broadening this thread in the discussion. Also, since this is such an important observation, I would suggest moving the supplemental tables 1-2 to the body of the text, which will make readers pay more attention to it.

Author Response

The article is very interesting and presents well-planned research. I have 3 comments on how to present the results:

Response: Thank you for your comments!

  1. The authors mention that the studies were conducted during Covid-19. Could this circumstance and possible changes in the course of examinations of the individual health condition or health of family members modify the obtained results, especially those concerning the level of anxiety, depressive symptoms, and daily stress?

Response: Since the study was a randomized controlled trial, it is unlikely that Covid-19 had impact on observed associations. Covid-19 had effects on the level of anxiety, depressive symptoms, and daily stress in both intervention and control groups. It is unlikely that Covid-19 had impact only on control or intervention group.

  1. I would suggest a clearer comparison of the effects of individual and group lifestyle coaching as a guide when planning future research.

Response: Unfortunately, this trial had low statistical power to compare individual and group coaching, particularly the sample size was small for individual coaching. The findings are not robust to conclude that there is a clear difference between these two types of coaching.

  1. The authors point out that the lifestyle changes demonstrating the success of the intervention were mainly seen among workers with fewer overlapping tasks, less busy schedules, or less work-intensive work. Hence, it is correct to conclude that it is advisable to modify the resources of employees at work so that they can benefit from coaching and other methods used in lifestyle interventions. I would suggest broadening this thread in the discussion. Also, since this is such an important observation, I would suggest moving the supplemental tables 1-2 to the body of the text, which will make readers pay more attention to it.

Response: We have moved supplemental tables 1-2 to the body of the text as Tables 4-5 and discussed further this issue.

Reviewer 3 Report

Review for Manuscript entitled “The Effect of Healthy Lifestyle Changes on Workability and 2 Wellbeing: A Randomized Controlled Trial”.

Thank you for providing me with the opportunity to review the manuscript entitled “The Effect of Healthy Lifestyle Changes on Workability and 2 Wellbeing: A Randomized Controlled Trial”.

Title:

The title of the manuscript does not reflect the study performed. It talks about a randomized clinical trial and the randomization is not correct.

Introduction:

The aim of the study is not correlated with the Results section and the Conclusions section of the study. Stress is not mentioned in the objective, but it is in the results.

Materials and Methods:

What inclusion criteria have been established for the design of this study? This have to be explained and added. In the Participants section, only the following are mentioned healthy workers aged 18-65 years who were able to speak Finnish or English

In the study there is no randomization in the assignment of individual or group coaching. The manuscript says that according of ApoB/ApoA1 ratios, it was assigned to one type of coaching. In the Randomization section it is said that there is no blinding.

Outcome measures are poor. Validated scales that provide more useful information should be included.

Results:

The sample included is not homogeneous. There is a big difference between the individual coaching group and group coaching.

There is a significant difference between the gender of the sample.

Conclusion:

The aim of the study is not correlated with the Conclusions section of the study.

Author Response

Thank you for providing me with the opportunity to review the manuscript entitled “The Effect of Healthy Lifestyle Changes on Workability and 2 Wellbeing: A Randomized Controlled Trial”.

Response: Thank you for your comments!

Title:

The title of the manuscript does not reflect the study performed. It talks about a randomized clinical trial and the randomization is not correct.

Response: We have modified the title: “the effect of healthy lifestyle changes on workability and mental health symptoms: a randomized controlled trial”. This is a randomized controlled trial with four arms. A proper randomization method was used to randomize the participants. The characteristics of intervention and control groups are balanced at baseline. The individual and group lifestyle coaching arms are independent groups because they had their own control group.

Introduction:

The aim of the study is not correlated with the Results section and the Conclusions section of the study. Stress is not mentioned in the objective, but it is in the results.

Response: We have reported the results for the aim of the trial. In the results section, we have reported the findings on the associations of lifestyle coaching with workability, mental health, and sleep. Additionally, we have reported results for possible effect modifiers of the observed associations.

Materials and Methods:

What inclusion criteria have been established for the design of this study? This have to be explained and added. In the Participants section, only the following are mentioned healthy workers aged 18-65 years who were able to speak Finnish or English

Response: We have listed other inclusion criteria on page 2, last paragraph.

In the study there is no randomization in the assignment of individual or group coaching. The manuscript says that according of ApoB/ApoA1 ratios, it was assigned to one type of coaching. In the Randomization section it is said that there is no blinding.

Response: Randomization prevents selection bias, while blinding prevents detection bias (measurement of outcomes). Blinding is not feasible for some interventions. It is not possible to blind participants to coaching. We have used a block randomization to balance participants according to their level of cholesterol. Random blocks also create successful randomization.

Outcome measures are poor. Validated scales that provide more useful information should be included.

Response: This trial has already been completed and measurement methods cannot be changed or validated. We have used a validated tool for measuring workability. Previous studies have shown that two questions are enough to screen for depressive and anxiety symptoms. We have discussed the limitations of our outcomes’ assessment.

Results:

The sample included is not homogeneous. There is a big difference between the individual coaching group and group coaching.

There is a significant difference between the gender of the sample.

Response: There were no differences in baseline characteristics between group lifestyle coaching and its control group. There were two differences in baseline characteristics between individual coaching and its control group. These two differences were borderline statistically significant. Since the sample size of individual coaching and its control group was small, random variation could create imbalances at baseline.

Conclusion:

The aim of the study is not correlated with the Conclusions section of the study.

Response: The conclusions of the study are in line with the aim of the study. The conclusions also include some recommendations for future research on this topic.

Round 2

Reviewer 1 Report

Response: The primary objective of this trial was to study the effect of lifestyle coaching on blood metabolite levels. Cholesterol metabolism, blood glucose, triglycerides, low grade inflammation, amino acid metabolism and purine metabolism were measured. The secondary objective was to study the effect of lifestyle coaching on workability, mental health symptoms and sleep. We have now explained this on page 3, 2nd paragraph.

While this addition helps a bit, the results of your primary objective are not reported in Results. And the results of the secondary objectives are not put into any context with the "high-risk" and "low-risk" groups you mention in the methods. e.g. Where in the manuscript can the reader find the effects of lifestyle coaching on blood metabolite levels? Are there differences between coaching outcomes for your secondary objectives in the high risk versus low risk groups? Is this allocation to high risk and low risk groups of any significance to your secondary objectives at all ?

Response: For the job strain, only information on overlapping jobs, tight schedules and stretching workdays was collected. The use of those items in job strain measure can be justified by a study population. Most of the participants worked in knowledge jobs. Overlapping jobs, tight schedules and stretching workdays are typical in these jobs. https://link.springer.com/article/10.1007/s41542-020-00058-1

Excellent; consider adding this explanation and reference to the manuscript for the reader to better follow your methodological approach.

Minor: Discussion "One of the reasons for the absence of effects on anxiety and sleep in the current trial might be due to the short follow-up period."

Author Response

Thank you for your comments!

1) The primary outcomes have been reported in another paper. However, the paper has not yet been published. The low-risk and high-risk groups were defined based on the serum lipids. Reporting the results for low-risk and high-risk groups is not applicable to the secondary outcomes.

2) We have added the explanation to the manuscript and cited the reference.

3) We have added “the” to the sentence.

Reviewer 3 Report

After the revision, the manuscript is ready to be published. Congratulations

Author Response

Thank you for your positive comments!